# Technological Assessment on Steam Reforming Process of Crude Glycerol to Produce Hydrogen in an Integrated Waste Cooking-Oil-Based Biodiesel Production Scenario

**Vishal Naranje** [1,*] , **R. Swarnalatha** [2,*] , **Ojas Batra** [1] **and Sachin Salunkhe** [3]

1    Department of Mechanical Engineering, Amity University Dubai, Dubai International Academic City, Dubai P.O. Box 345019, United Arab Emirates
2    Department of Electrical and Electronics Engineering, Birla Institute of Technology & Science, Pilani, Dubai Campus, Dubai International Academic City, Dubai P.O. Box 345055, United Arab Emirates
3    Vel Tech Rangarajan Dr. Sagunthala R&D Institute of Science and Technology, Chennai 600062, India
*    Correspondence: vnaranje@amityuniversity.ae (V.N.); swarnalatha@dubai.bits-pilani.ac.in (R.S.)

**Abstract:** The current scenario of society is to produce fuel from renewable energy resources. The purpose of this research work is to develop an integrated approach for glycerol valorization and biodiesel production. Employing a range of methodologies widely used in the industry, technical analysis and assessments of the process's applicability in real-world situations are also made. The integrated process plant is simulated using Aspen Plus®. Several different sensitivity analyses are carried out to describe the process that improves efficiency and are designed to maximize hydrogen recovery from the reforming section. The integrated process results are compared with several existing standalone biodiesel production processes. Additionally, the results are verified with the theoretical studies on glycerol valorization. The outcomes of the process plant simulation reveal coherent results with the current industrial standards for the two processes. The results show that the amount of glycerol produced (stream 7) is 60.72 kmol/h in mass flow rate, this translates to 7272.74 kg/h. The hydrogen produced is 488.76 kmol/h and, in mass flow rate, this translates to 985.3 kg/h. The total yield of hydrogen produced is around 13%. The biodiesel yield is at 92.5%. It shows a realistic recovery that would be attained if the process is implemented, contrary to theoretical studies.

**Keywords:** biodiesel production; glycerol valorization; hydrogen production; sensitivity analysis; integrated process plant

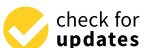



## 1. Introduction

Fossil fuel depletion worldwide has urged people to look for alternative energy sources to decrease the use of depleting resources [1]. In addition to the strong dependence of people on naturally occurring fuels, there is also a proven direct negative impact of these on the environment, mainly the release of carbon dioxide into the environment, which is estimated to be a massive 21.3 billion tons per year [2]. All these negative impacts and dependency have caused an increased focus on using alternative energy sources such as wind, geothermal, solar, biofuels, and many more. As there has been a very driven approach toward creating renewable solutions and reducing the dependency on fossil fuels, progress has been made in countries such as the US, where 20% of the energy produced is now coming from renewable sources [3]. One of the main applications of fossil fuels in the modern world is diesel fuel. It is one of the most heavily used fuels mainly in transport and energy generation [4]. Its price and efficiency in engines made it the best option for consumers and large-scale transport applications such as public transport, trucks, and even freight [5]. However, as with other products of fossil fuels, resources continue to lessen, and alternatives must be found. One such alternative to the extensively used diesel is biodiesel.

Biodiesel is a product derived from plant and animal oils. It is formed mainly and widely through transesterifying these fats and oils with alcohol to form methyl esters and glycerol. The methyl ester is what can then be used as a substitute for conventional diesel. The byproduct of biodiesel production, glycerol, is produced at a ratio of 9:1 for methyl ester to glycerol, respectively [6]. With increasing biodiesel production worldwide, the handling of low-purity glycerol has become a growing concern with producers. Many different options exist for dealing with it, from purification to being used as an energy source. Some methods for glycerol-free production of biodiesel have also been proposed [7–11]. Biodiesel is a monoalkyl ester obtained from either plant, animal, or waste oils. Over the past two decades, it has been studied extensively to analyze its credibility as an alternative to common crude-oil-derived diesel fuel [12–15]. The various literature revealed that there needs to be no modification in most existing diesel engines to apply biodiesel in them. In addition to being a good substitute for diesel, it is considered to be better than its toxic predecessor in many ways, such as being non-toxic, biodegradable, a better lubricant, and carbon-neutral [16]. Even though biodiesel is a viable replacement for conventional diesel, it is often criticized as a possible cause of imbalance in the global food demand and supply market and the inflation and shortages of food products. A total of 34% of all edible oil produced worldwide was used in biodiesel production between 2004 and 2007. These concerns are raised due to its source being edible oils. However, even non-edible-oil-derived biodiesel has been criticized for increasing competition toward arable land and water resources [17]. Waste cooking oil remains a viable and largely uncriticized raw material for biodiesel production. It is described as a third-generation feedstock in addition to microalgae and animal fat [18]. They have been put in this category because they present some to no market value and would otherwise go to waste. An interesting fact also considered is that feedstock price usually accounts for over 80% of the actual production cost of biodiesel. Therefore, the biodiesel industry itself has been searching for cheaper alternatives and sustainable sources [19]. In a study, the authors investigated the use of a clay/CaO heterogeneous catalyst for the production of biodiesel from waste cooking oil [20]. In a study, the experiments were designed via the Box–Behnken method and experiments were performed to investigate and optimize the effects of the variables of calcined-cow-bone-to-KOH ratio, oil-to-methanol volume ratio, residence time, and reaction temperature on the purity of biodiesel [21]. This study offered an overview of the latest advances in the design of graphene-based materials for delivery of bioactive agents [22]. In the study, composites of CaO and MgO were used for producing biodiesel from waste cooking oil, and its efficiency was studied under optimum conditions [23].

The present investigation deals with the conversion of glycerol to hydrogen through different methods that have been widely researched throughout the industry. One of those methods, steam reforming, is investigated and simulated in this research to extend a generic biodiesel production process. This simulation provides insight into the recoveries that can be obtained by simulating an integrated plant. The description given in this study has a novel approach. Several enhancements are made to the plant to optimize the hydrogen yield from the reforming process discussed. Finally, the feasibility and applicability of the research are compared to other works carried out in the same field.

## 2. Crude Glycerol

Crude glycerol usually has a purity ranging from 40 to 85%, depending on which feedstock was used to produce the biodiesel [24]. Biodiesel production facilities will need to install equipment accordingly to deal with the quality of glycerol that is being attained with their respective feedstocks. There are several different options companies utilize to deal with the glycerol that is produced as a byproduct. One of the more obvious methods is purifying the crude glycerol to be sold to be used as a raw material. Glycerol, with high purity, is a utility with applications in many fields, and its non-toxicity to humans also opens it to several industries such as food and medicine [25,26].

Several studies have been carried out on the purification and sale of glycerol derived from biodiesel production. It is well known that the purifying process depresses the price of glycerol, which has a negative impact on the market. This is a result of the market becoming oversaturated. The current market value of pure glycerol is USD 0.27–0.41 per pound; however, the crude glycerol with 80% purity is as low as USD 0.04–0.09 per pound. This proved that excessively produced glycerol affects the price of the glycerol in the market. Therefore, utilization of the crude glycerol for value-added products has become a serious issue in the biodiesel industry [27].

The supply drivers of glycerol production changed from being a product of mainly fatty acids and soap manufacturing processes to being a byproduct of the renewable energy industry. In 1999, only 9% of the total glycerol produced worldwide was from the biodiesel production process. In 2009, the number rose to 64%, representing the large impact of the industry. The prices of crude glycerol continue to decrease because of the increasing biodiesel production and negative impact on the overall biodiesel production process [28]. The price of crude glycerol is continuously dropping in the market as biodiesel production is unceasingly increasing [29]. Crude glycerol production was projected to rise to 5.8 billion pounds in 2020, presented largely by the European Union, as shown in Figure 1. Purifying crude glycerol to a higher-purity product and supplying it to markets that use it seems to be an economically less suitable prospect from the discussion above. These data have contributed highly to increased research in exploring other methods of utilizing the crude glycerol produced from the process.

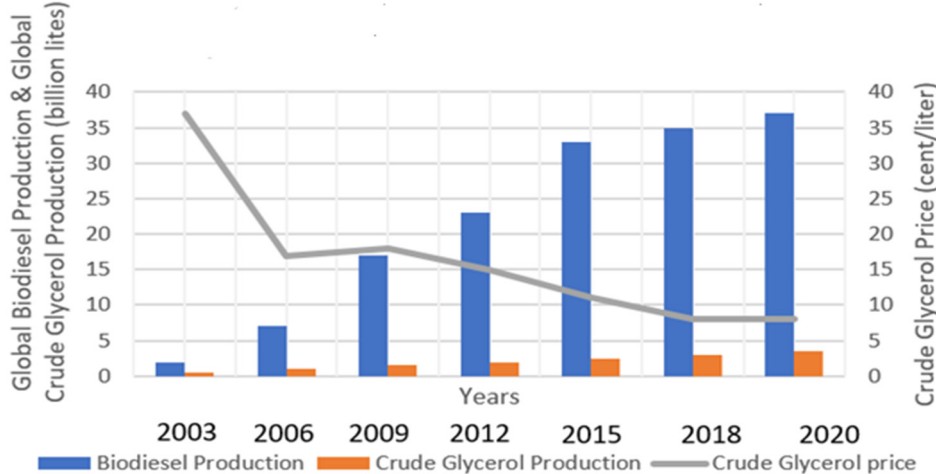

**Figure 1.** Global biodiesel production of crude glycerol production [30].

Several other methods of utilizing crude glycerol and producing different products exist. Table 1 summarizes the several different methods of production and final products that can be produced from crude glycerol [31].

**Table 1.** Different processes for utilization of crude glycerol.

| Process Pathway | Final Product |
|---|---|
| Purification | Glycerol |
| Chlorination | Dichloro-propanol and epichlorohydrin |
| Steam Reforming | Hydrogen and syngas |
| Dehydration | Acrolein, acrylic acid, and acetol |
| Hydrogenolysis | Propanediol, ethylene glycol, propanol |
| Oxidation | Glycolic acid, formic acid, and other acids |
| Esterification | Glycerol, tertiary butyl ether, and other ethers. |
| Transesterificatin | Glycerol Carbonate and Methanol |

### 3. Crude Glycerol to Hydrogen

As shown in Table 1, hydrogen is one of the final products of crude glycerol valorization. On its own, as an alternative for crude glycerol usage, hydrogen has been studied extensively for different conversion methods. In addition to its numerous industrial applications, what makes hydrogen an attractive choice is the current production method for most hydrogen worldwide. A total of 95% of hydrogen production around the world is fossil-fuel-based. Therefore, producing it from renewable resources such as biodiesel presents a great opportunity to reduce fossil fuel consumption [32]. From a variety of household sources, including biomass, fossil fuels, and water electrolysis with electricity, hydrogen can be produced. With the development of innovative, environmentally friendly techniques, the produced hydrogen can be used as a source of energy for its production plant without the need for storage or transit [33].

The crude glycerol is converted into hydrogen based on biological, water electrolysis, and thermochemical processes. The biological processes present are fairly new and hold potential because the reactors are at ambient conditions, requiring very little external energy [34]. Hydrogen production from water electrolysis is a specialized process that produces high-purity hydrogen. This process is, therefore, very expensive because of specialized equipment as well as a very niche market. For these reasons, it is not used commonly [35]. The thermochemical processes for hydrogen production are the ones that are most popular in the industry and highly investigated. Even though these processes are largely endothermic, they present high conversion and efficiency rates [36].

The different thermochemical processes for glycerol production are steam reforming (SR), AutoThermal reforming (ATR), SuperCritical Water Reforming (SCWR), Partial Oxidation Reforming (POR), and Aqueous Phase Reforming (APR).

The SR method is the most commonly used method for hydrogen production from $CH_4$. This method consists of two main reactions. The first is the global reaction of steam reforming glycerol over a catalyst, producing syngas of H, CO, and CO. There are also parallel methanation reactions that convert carbon dioxide and monoxide to produce methane and water [37]. In the study, the authors carried out a thermodynamic analysis of the SR process by varying temperatures and pressures. For maximal hydrogen production, it is recommended to operate the process at 625 °C temperature and 1 bar pressure [38].

The POR method is governed by the glycerol oxidation reaction. At 1 atm pressure, glycerol reacts with oxygen to produce syngas as steam reforming CO, $CO_2$, and H2 [39]. However, several parallel reactions can be carried out due to how rapid the consumption of oxygen is, including the oxidation of glycerol to CO, HO, and not H, and CO and H and not CO. The ATR method combines the SR reactions and the POR reactions, both the global steam reforming reaction described previously, and the three partial oxidation reactions for POR in the ATR reactor. The ATR reaction is more researched because of its quality of energy efficiency because of one endothermic and several exothermic reactions occurring in the reactor. Several thermodynamic and catalytic studies on this process indicate that the optimum temperature for the operation of an ATR reaction is between 625 °C and 725 °C and that at higher pressures, there was a higher conversion of glycerol and, consequently, a lower production of hydrogen [40,41]. In addition, the catalyst promoted by calcium was the scenario in which maximum conversion and hydrogen selectivity were achieved [42].

The APR method starts with glycerol decomposition, where the crude glycerol is converted to hydrogen and carbon monoxide and is followed by a water–gas shift reaction, which converts the CO into hydrogen and carbon dioxide using water; these occur at about 240 °C and 42 bar [43]. The reaction occurs in a liquid phase and has advantages over a traditional SR system, such as much greater heat recovery due to the phase and lower operating temperatures for the column [44]. However, they have major disadvantages and challenges such as lower selectivity of the hydrogen reaction over the methane reaction. This process is still relatively new and has to overcome the disadvantages compared to more traditional processes.

Finally, the SCWR process is an alternative route for hydrogen production that is carried out in the critical conditions of the water. After a detailed thermodynamic analysis, in the study, authors attained the optimum conditions for this reaction to be 240 atm and the temperature to be between 750 °C and 800 °C [45]. This process contains many features such as reduced catalyst requirement, and 100% glycerol conversion can be achieved. It makes this process an attractive opportunity for the valorization of crude glycerol, demonstrated by a techno-economic analysis carried out in the study [46]. The authors also studied the application of an SWCR process in 3 different biodiesel production scenarios and several other studies.

## 4. Process Simulation

### 4.1. Standalone Biodiesel Production Process

The biodiesel production process simulated for this research work is based on a 2-step process by [47]: to convert high-FFA-content fatty acids into methyl esters, i.e., biodiesel. To make the simulation realistic, the same composition was used for the feed. The biodiesel process itself is a highly simulated one and the purpose of this study is to build a much bigger integrated plant simulation that can then be used for future work such as economic or energy or other such further analyses on the feasibility of the plant. The 2-step process is mentioned as follows:

Step 1: Hydrolysis:

$$C_{57}H_{104}O_6 + 3H_2O \quad C_3H_8O_3 + 3C_{18}H_{34}O_2 \tag{1}$$

Triolein + Waler Glycerol + Oleic Acid

$$C_{57}H_{98}O_6 + 3H_2O \quad C_3H_8O_3 + 3C_{18}C_{32}O_2 \tag{2}$$

Trilinolein + Water Glycerol + Linoleic Acid

These reactions take place in the first reactor, a stoichiometric reactor at 11 MPa and 290 °C, as recommended in [48]. The reason this reactor was chosen is that the hydrolysis reactions are reversible and so the accumulation of the product is required. The 2 streams entering are first brought to that pressure by pumps and then the hydrolysis reaction occurs. The products of these streams are then sent into a distillation column that removes most of the glycerol and water as the distillate, and the remaining triolein and trilinolein produced oleic acid and linoleic acid and the rest of the water is removed as the bottom product. The conversion of the oil to the acids by hydrolysis reaction in current industrial practice is around 96–99%; 96% conversion is used for this simulation [49]. The bottom stream containing a combined acid mole fraction of 0.835% is then sent to the second reactor for the second reaction, esterification.

Step 2: Esterification:

$$C_{18}H_{34}O_2 + CH_3OH \quad C_{19}H_{36}O_2 + H_2O$$

Oleic Acid + Methanol Methyl Oleate + Water  (3)

$$C_{18}C_{32}O_2 + CH_3OH \quad C_{19}H_{34}O_2 + H_2O$$

Linoleic Acid + MethanolMethyl Linoleate + Water  (4)

These reactions are carried out in a continuously stirred tank reactor recommended by several sources for esterification [50,51]. As these reactions are carried out in a CSTR, they must be specified more than the ones in a batch reactor. These specifications are the thermodynamic and kinetic data, i.e., the reaction rate constant and the frequency factor. These values were obtained from previously reported experimental data to be 0.0006 s$^{-1}$, 50.5 kJ/mol, and 4.05 s$^{-1}$ [52]. The methanol-to-oil ratio is one of the most important things in the design of this reactor mainly because most of the methanol is recycled back to the reactor. The fresh methanol-to-oil ratio was calculated by sensitivity analysis to be 2:1 for

maximum conversion. The rest of the process is basically separation and further refining to give a product with a high methyl ester mole fraction as well as recycling of the methanol, as shown in Figure 2.

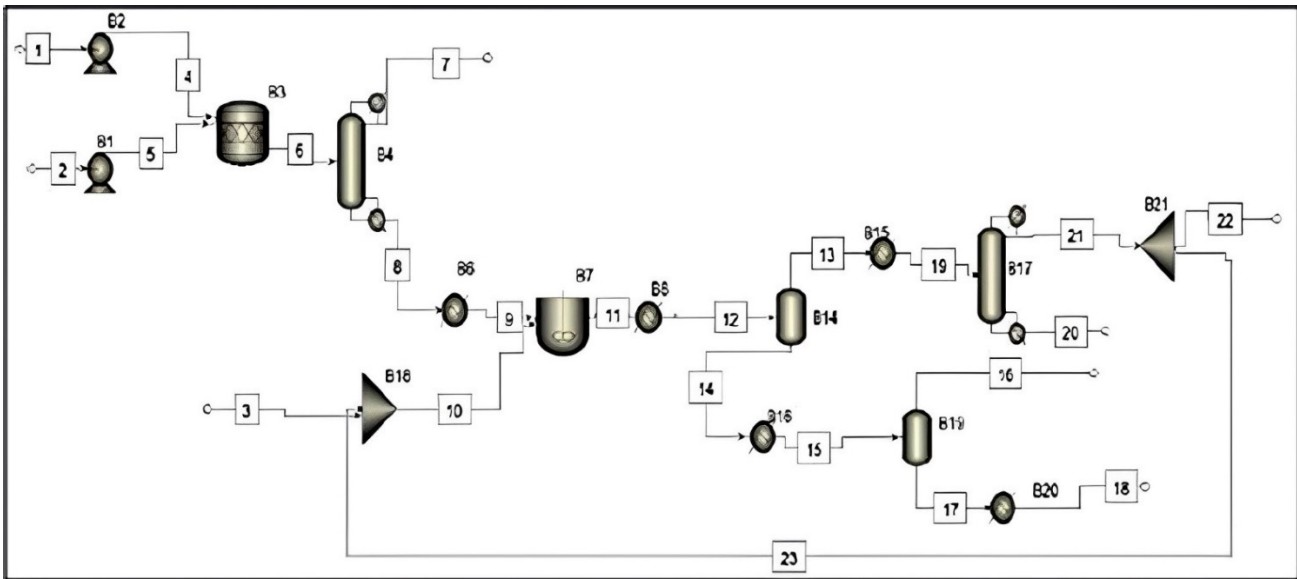

**Figure 2.** Biodiesel Production Process.

B3 is the hydrolysis reactor. B4 is the distillation column with its top stream, 7, being the crude glycerol stream. The bottom product is carried on to the esterification reactor B7 where the methyl ester is formed by the reaction of acids with methanol.

### 4.2. Crude Glycerol Valorization Process and the Process Integration

This glycerol process is based on widely used techniques for the steam reforming of glycerol. Before the hydrogen separation process begins, a flash drum is used to flash the water out of the crude glycerol stream. Pyrolysis, water gas shift, methanation, and steam reforming of methane are the four desired reactions that need to occur inside Gibbs reactors. These reactors calculate reactor outlet temperatures by minimizing the Gibbs free energy, which limits CO production [53].

Pyrolysis:

$$C_3H_8O_3 \leftrightarrow 3CO + 4H_2$$

Glycerol ↔ Carbon Monoxide + Hydrogen

Methanation Reaction:

$$CO + 3H2 \leftrightarrow CH4 + H2O$$

Carbon Monoxide + Hydrogen ↔ Methane + Water

Water-Gas Shift reaction:

$$CO + 3H_2O \leftrightarrow CH_4 + 2HO$$

Carbon Monoxide + Hydrogen ↔ Methane + Water

Steam Reforming of Methane:

$$CH_4 + H_2O \leftrightarrow CO + 3H_2$$

Methane + Water ↔ Carbon Monoxide + Hydrogen

It can be seen that the methanation and water–gas shift reactions are driving the methane production, which is then reformed with steam to produce carbon dioxide and hydrogen; the direct pathway for hydrogen production is pyrolysis. Mainly pyrolysis takes

place in the first Gibbs reactor simulated in this research. The rest of the reactions occur in the second Gibbs reactor. It creates a large possibility for coke formation in the second Gibbs reactors. The excess steam is used for the reforming reaction so that all the methane would have thermally decomposed instead of reacting with water first. The Gibbs reactors work well in reducing the overall COx production as mentioned before, limiting the level of coke that can be formed using CO decomposition and the coke formation reactions that CO and $CO_2$ undergo with hydrogen.

The hydrogen produced is separated from the gas mixture using a technique called pressure swing adsorption. This technique is highly used in industries for hydrogen separation and proves to be quite an efficient technique for producing hydrogen at specified purities, sometimes up to 99.95% volume [54]. The technique consists of a packed bed reactor with strippers that uses micro and mesoporous adsorbents to strip the gas. A simple version of this technique is simulated on Aspen Plus® using the 'Sep' feature, a piece of equipment that separates an inlet stream based on the splits of each component required. The split specified for the stream exiting the pressure swing reactor is set to 90% hydrogen.

Figure 3 shows the flow diagram of the integrated process for biodiesel production and glycerol reforming to hydrogen. A blue box in Figure 3 indicates the following.

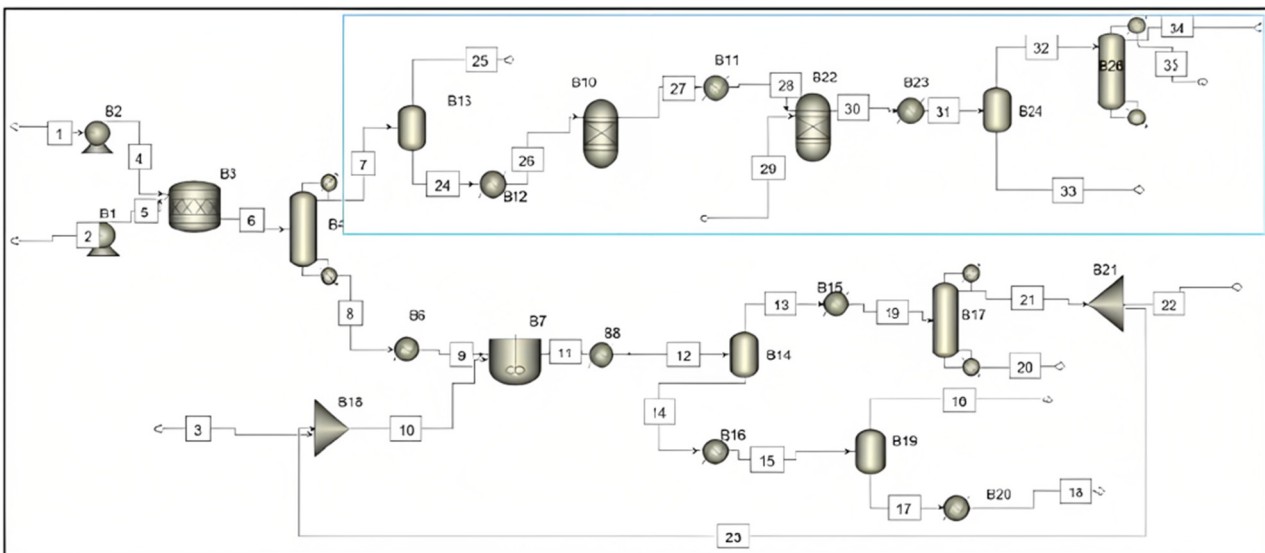

**Figure 3.** Flow process diagram of the Integrated Process.

Stream number 7 is the crude glycerol stream from which water becomes separated in flash drum B13. The bottom product is sent into the pyrolysis reactor B10, then it is sent into the steam fanning reactor, B22. After the liquid components of the resulting streams are knocked out in flash drum B24, pressure swing adsorption is carried out in equipment ID B26, and the 90% pure hydrogen stream number 34 is the final product of the extension.

## 5. Result and Discussion

The integrated process of biodiesel production and glycerol reforming needs to be optimized for maximizing hydrogen production from glycerol.

Block number B4 is a separation column used to separate the hydrolysis reaction products, the crude glycerol, and some water leaves as the distillate. In contrast, the bottoms product is mainly the acids. An insignificant amount of unreacted triolein and trilinolein is also present in both streams, but this amount is very insignificant due to the high conversion rate of the hydrolysis reaction. This preliminary distillation is crucial as it provides the feed for the steam reforming process for hydrogen production. It needs to be ensured that maximum separation of the crude glycerol and acids takes place. The first sensitivity analysis was carried out on distillation block B7, and the results are shown in Figure 4a,b.

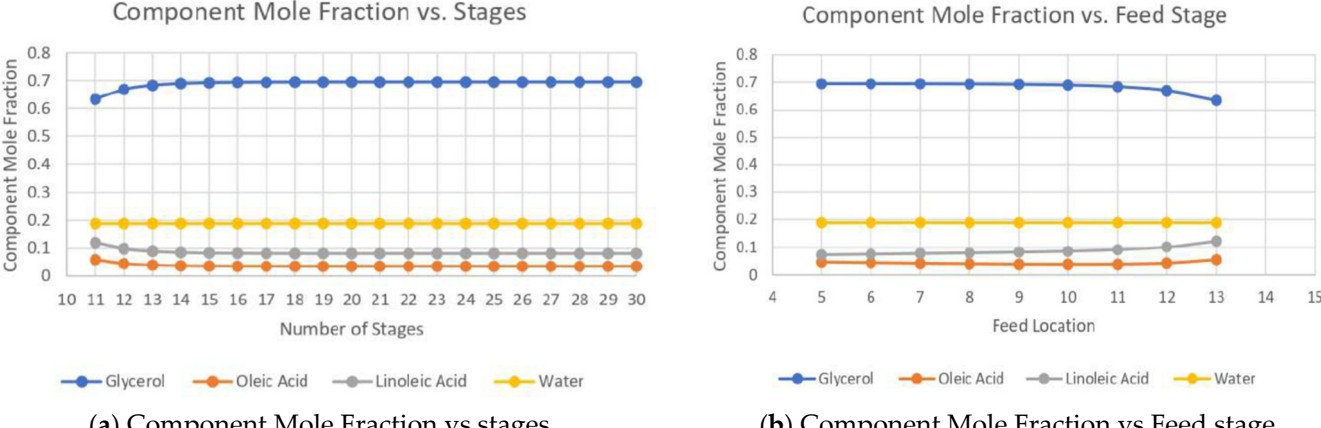

(**a**) Component Mole Fraction vs stages

(**b**) Component Mole Fraction vs Feed stage

**Figure 4.** Sensitivity Analysis on distillation block B7.

The mole fractions represented in these figures are of the crude glycerol stream, 7. It can be deduced from Figure 4a that, after 15 stages, the amount of glycerol in the top stream has reached its maximum mole fraction, and the acids have reached their minimum value. Similarly, for the second sensitivity analysis, the best feed location is determined to be 5 as shown in Figure 4b.

*Analysis of Gibbs Reactor for Steam Reforming, B22*

The goal of the Gibbs reactor placed in this simulation is used to maximize the amount of hydrogen in the product stream 30. The data from the sensitivity analysis help to decide the stream amount. The mole flow of the hydrogen in kmol/h is the clear choice to define for the analysis, but it is also important to make sure that the mole fractions are consistent with the increase or decrease in mole flow. The two critical parameters for hydrogen production are the reactor's temperature and the flow of steam going into it. Figure 5a,b represent results from the sensitivity analysis.

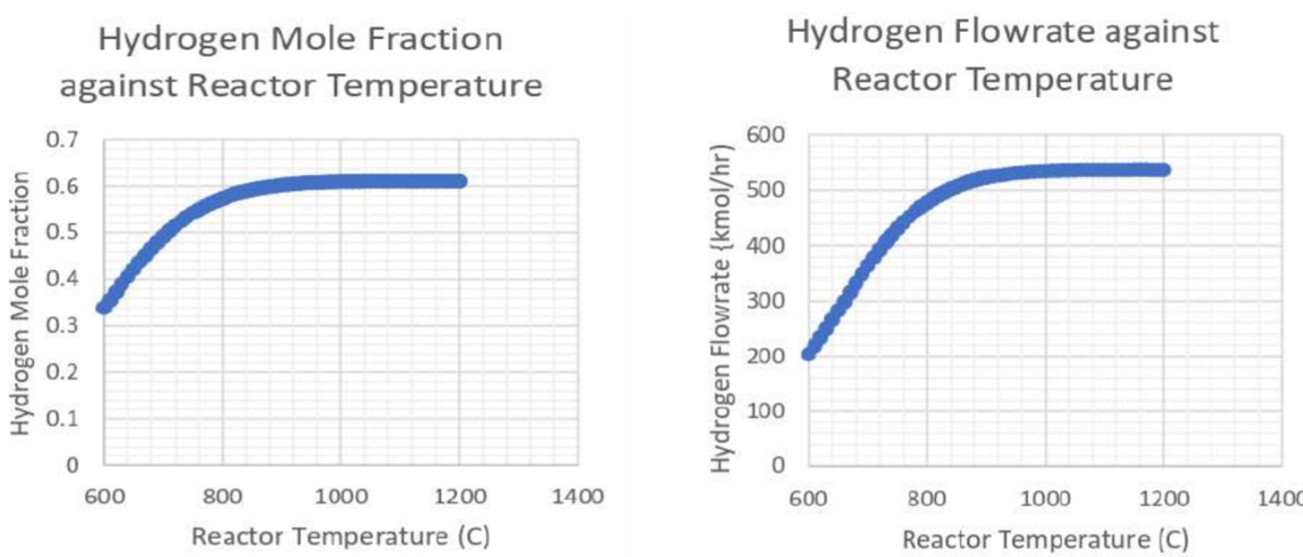

(**a**) Hydrogen Mole Fraction vs Reactor Temperature

(**b**) Hydrogen Flow rate vs Reactor Temperature

**Figure 5.** Sensitivity Analysis of Steam Reforming block B22.

The relationship between Hydrogen mole fraction against reactor temperature is shown in Figure 5a. The result of Hydrogen Flow rate against reactor temperature is shown in Figure 5b. Both findings demonstrate that the graphs are steady and rising

concurrently. With this temperature, both the parameters plateau at 910 °C. Therefore, the reactor temperature needs to be set at this value.

　　Figure 6 displays the outcomes of varying the steam flow rate in accordance with the mole flow and hydrogen mole fraction in stream 30. The hydrogen mole flow and mole fraction are consistent with each other after a certain value, as shown in Figure 6a,b. At 180 kmol/h, the hydrogen flow rate increases at a much slower rate than it was doing before, and at the same point, the mole fraction starts dropping as opposed to increasing before. Not only does this help us pinpoint what the optimum steam flow rate is at that temperature, but it also reveals a very interesting fact about the reactions involved in the Gibbs reactor. As shown previously, the reactions in the second Gibbs reactor are reversible, and the sensitivity analysis indicates the boundary flow rate for the reverse reaction to occur. When the rate of steam entering the reactor exceeds 180 kmol/h, the reverse reaction is sparked because of the hydrogen saturation in the reactor. It then begins reacting with CO and $CO_2$, which leads to coke formation. The produced hydrogen is used up internally instead of being carried forward to the product stream.

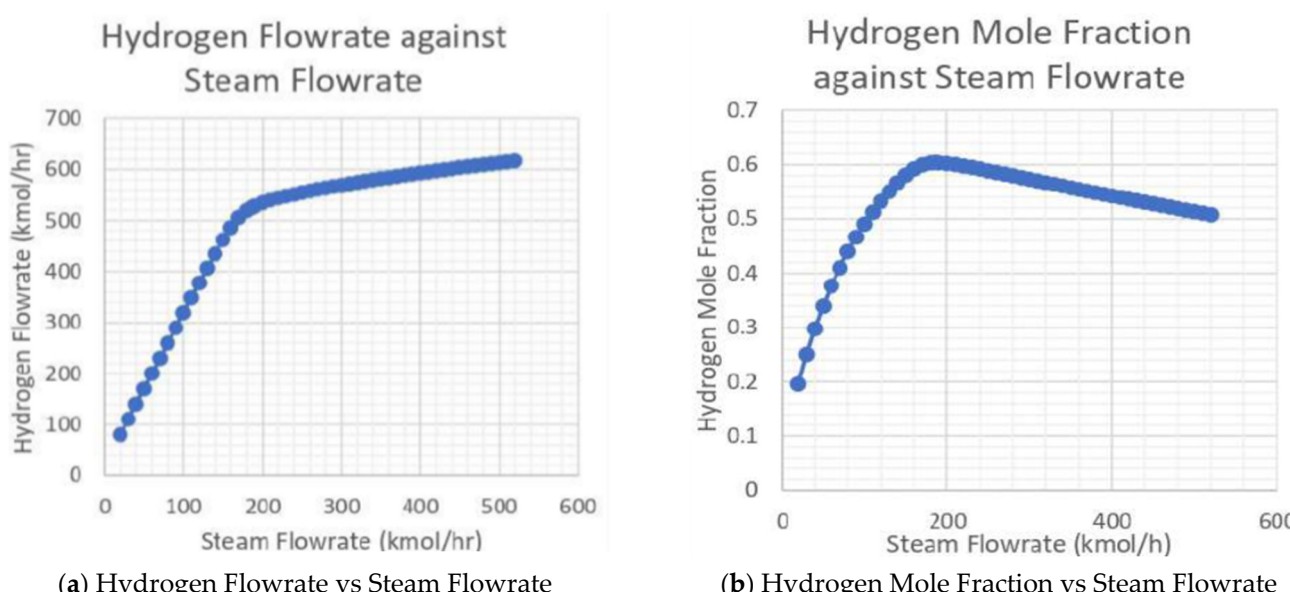

(**a**) Hydrogen Flowrate vs Steam Flowrate　　　　　(**b**) Hydrogen Mole Fraction vs Steam Flowrate

**Figure 6.** Sensitivity Analysis of steam flow rate.

　　Table 2 presents the main keystream results of the integrated process. These stream results show the water in, waste cooking oil in, methanol in, biodiesel out, pyrolysis outlet stream, second Gibbs reactor outlet stream, and the hydrogen-produced stream. The mole fractions for the inlet streams, which have only one component, have not been listed, because they are redundant. Streams 1, 2, and 3 for water, waste cooking oil, and methanol inlets, respectively, need to be analyzed for this integrated simulation approach. The amount of biodiesel produced when compared to the waste cooking oil also needs to be evaluated. The biodiesel product stream, 18, has a combined 80% of biodiesel components, methyl-oleate and methyl-linoleate. Compared with the results of recent literature that showed new and advanced technologies for biodiesel production from waste cooking oil, this value is quite low [23]. The method chosen for this production process was limited to a certain purity of the final biodiesel, and this was chosen on purpose so that the absolute base scenario can be represented [55]. The single-step transesterification approach is more common with research that presents much higher purities of biodiesel.

**Table 2.** Key Stream Results for the Integrated Process.

| Stream | 1 | 2 | 3 | 7 | | 18 | | 27 | | 30 | | 34 | |
|---|---|---|---|---|---|---|---|---|---|---|---|---|---|
| Temperature (°C) | 25 | 25 | 25 | 219.3189 | | 25 | | 732 | | 25 | | 25 | |
| Pressure (bar) | 1.01325 | 1.01325 | 1.01325 | 1.01325 | | 1.01325 | | 1.01325 | | 1.01325 | | 1.01325 | |
| Component: | Kmol/h | Kmol/h | Kmol/h | Kmol/h | Mole Fraction | Kmol/h | Mole Fraction | Kmol/h | Mole Fraction | Kmol/h | Mole Fraction | Kmol/h | Mole Fraction |
| Water | 138 | 0 | 0 | 5.52 | 0.090909 | 8.024471 | 0.060308 | $5.42 \times 10^7$ | $1.64 \times 10^9$ | 12.08372 | 0.030595 | 0 | 0 |
| Triolein | 0 | 23 | 0 | $1.47 \times 10^9$ | $2.43 \times 10^{11}$ | 0.92 | 0.006914 | 0 | 0 | 0 | 0 | 0 | 0 |
| Methanol | 0 | 0 | 190 | 0 | 0 | 1.903104 | 0.014303 | $5.55 \times 10^{13}$ | $1.68 \times 10^{15}$ | $5.14 \times 10^7$ | $5.51 \times 10^{10}$ | 0 | 0 |
| Oleic-acid | 0 | 0 | 0 | 4.37255 | 0.072012 | 5.311773 | 0.039921 | $4.32 \times 10^{10}$ | $1.30 \times 10^{12}$ | 0 | 0 | 0 | 0 |
| Glycerol | 0 | 0 | 0 | 44.15232 | 0.727146 | 0.007627 | $5.73 \times 10^5$ | 0 | 0 | $1.72 \times 10^{27}$ | $1.65 \times 10^{30}$ | 0 | 0 |
| Methyl-Oleate | 0 | 0 | 0 | 0 | 0 | 56.48453 | 0.424513 | $1.87 \times 10^{20}$ | $5.86 \times 10^{23}$ | 0 | 0 | 0 | 0 |
| Hydrogen | 0 | 0 | 0 | 0 | 0 | 0 | 0 | 0.000557 | $1.68 \times 10^6$ | 531.1917 | 0.602088 | 488.7636 | 1 |
| Methane | 0 | 0 | 0 | 0 | 0 | 0 | 0 | 181.6321 | 0.548213 | 2.994619 | 0.001438 | 0 | 0 |
| Carbon-monoxide | 0 | 0 | 0 | 0 | 0 | 0 | 0 | 139.2939 | 0.420424 | 322.6526 | 0.352064 | 0 | 0 |
| Carbon-dioxide | 0 | 0 | 0 | 0 | 0 | 0 | 0 | 10.38919 | 0.031361 | 5.668012 | 0.013815 | 0 | 0 |
| Oxygen | 0 | 0 | 0 | 0 | 0 | 0 | 0 | 0 | 0 | $3.58 \times 10^{17}$ | $2.04 \times 10^{19}$ | 0 | 0 |
| Trilinolein | 0 | 23 | 0 | $1.63 \times 10^9$ | $2.68 \times 10^{11}$ | 0.92 | 0.006914 | 0 | 0 | 0 | 0 | 0 | 0 |
| Linoleic-acid | 0 | 0 | 0 | 6.67513 | 0.109933 | 7.587828 | 0.057027 | $2.45 \times 10^{12}$ | $7.38 \times 10^{15}$ | 0 | 0 | 0 | 0 |
| Methyl-linoleate | 0 | 0 | 0 | 0 | 0 | 51.89777 | 0.390041 | $9.73 \times 10^{20}$ | $2.93 \times 10^{22}$ | 0 | 0 | 0 | 0 |
| Total | 138 | 46 | 190 | **60.72** | 1 | 133.0571 | 1 | 331.3158 | 1 | 874.5907 | 1 | 488.7636 | 1 |

The main comparison made in biodiesel production is the biodiesel produced rate compared to the total inlet of waste cooking oil. The total mass flow rate of the inlet waste cooking oil was 40,591.56 kg/h, and the biodiesel produced was 37,488.41 kg/h. It showed around 90% of biodiesel recovery, which was compared with the existing literature. The value was also nearer. The authors compiled a list of biodiesel yields from different sources, and it showed that the average value is about 92.5% [56]. These results are, therefore, coherent with real life.

The critical comparison of the research analyzed the hydrogen produced from the overall steam reforming process. The amount of crude glycerol produced (Stream 7) was 60.72 kmol/h. In mass flow rates, this translates to 7272.74 kg/h. The hydrogen produced was 488.76 kmol/h in mass flow rate, which translates to 985.3 kg/h. Therefore, the total yield of hydrogen from the crude glycerol was around 13%. One of the main challenges in hydrogen production from glycerol is the achievement of higher yield. There have been several advances in solving this challenge, but very few are ready for commercialization [37]. A compilation of recoveries achieved using the steam reforming method for the production of hydrogen from crude glycerol was studied, and it can be noticed that several processes showed recoveries of 3% [41]. The most prominent piece of research was the investigation of a commercialized Ni catalyst that helped achieve a yield of 70% [57]. Most of the processes that produce yields of above 70% are enhanced by high amounts of catalysts such as platinum, ruthenium, nickel, cerium, and other catalysts with rare compounds [41]. These must also be enhanced by other compounds, mainly oxides of metals. Theoretical comparisons showed that a lot of work needs to be carried out to develop technologies for crude glycerol reforming to hydrogen and their actual commercialization. As discussed before, several methods have been studied to increase hydrogen production especially focused on catalysts. Other studies considering reactors apart from Gibbs reactors have also been conducted.

## 6. Conclusions

In this research, the design and simulation of an integrated process for biodiesel production and glycerol valorization by conversion to hydrogen are being carried out using Aspen Plus® software. It is found that the biodiesel yield is at 92.5%, and the hydrogen yield attained from the reforming process is around 13%. The total mass flow rate of the inlet waste cooking oil is 40,591.56 kg/h, and the biodiesel produced is 37,488.41 kg/h. The amount of crude glycerol produced (Stream 7) is 60.72 kmol/h. In mass flow rates, this translates to 7272.74 kg/h. The hydrogen produced is 488.76 kmol/h. In mass flowrate, this translates to 985.3 kg/h. Therefore, the total yield of hydrogen from the crude glycerol is around 13%.

The results achieved for the yields of glycerol and hydrogen are proven to be coherent with several different theoretical and practical studies performed on the topic of both biodiesel production and crude glycerol valorization. These reveal the nature of the problem that currently surrounds the valorization aspect, achieving high yields for hydrogen from glycerol-reforming processes to be practically applicable.

This research work stands out and makes a substantial contribution to the field of study now being undertaken as it adopts an integrated approach to a subject that has historically only been explored individually. It provides a base for further studies that are more inclined toward applicability in actual industry and can even be further studied to and improved by heat integration, economic optimization, life cycle assessment, and several more techniques.

Biodiesel has been proven to be quite an efficient replacement as well as blend for already existing processes. In addition, the yields have been increased to an average of 90% for production processes around the world.

However, the utilization of its glycerol holds new challenges as simple purified glycerol does not hold much potential in the market due to saturation of supply. This research approach gives a potential solution to this issue by exploring one of the possible products

it can be turned into. The activity coefficients of reactions are taken into consideration during the simulation process. The sensitivity analysis reports that optimizing the performance of each equipment shows improved yield rates. The results achieved for the yields of glycerol and hydrogen are proven to be coherent with several different theoretical and practical studies performed on the topic of both biodiesel production and crude glycerol valorization.

**Author Contributions:** Conceptualization, V.N. and R.S.; methodology, V.N. and R.S.; software, O.B.; validation, R.S., V.N. and S.S.; formal analysis, V.N. and R.S.; investigation, R.S.; resources, R.S. and V.N.; data curation, O.B.; writing—original draft preparation, O.B.; writing—review and editing, V.N., R.S., O.B. and S.S.; visualization, R.S. and V.N.; supervision, R.S.; project administration, R.S., V.N., O.B. and S.S.; funding acquisition, V.N. and R.S. All authors have read and agreed to the published version of the manuscript.

**Funding:** This research received no external funding.

**Institutional Review Board Statement:** Not applicable, as no ethical approvals were required for this research work.

**Informed Consent Statement:** Not applicable.

**Data Availability Statement:** Not applicable.

**Conflicts of Interest:** The authors declare that they have no known competing financial interests or personal relationships that could have appeared to influence the work reported in this paper.

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
