# Peer review of "Technological Assessment on Steam Reforming Process of Crude Glycerol to Produce Hydrogen in an Integrated Waste Cooking-Oil-Based Biodiesel Production Scenario"

_processes, doi:10.3390/pr10122670_

Round 1

Reviewer 1 Report

Comments and Suggestions

1.     Introduction part is well written. The authors give a general introduction regarding biodiesel and its production and utilization.

2.     English grammar usage and scientific writing style need to be improved.

3.     Line 73-75: “This simulation gives insight into the recoveries that can be achieved by simulating an integrated plant, which is later in the paper described to be relatively rare”. Consider reconstructing the sentence to make the objectives of the paper clear.

4.     Line 88-89: “It is known that the purification process adversely affects the glycerol market by decreasing its prices due to its being an already saturated market.”: Consider rewriting the sentence. Check the usage of “due to”.  How do the purification processes adversely affect the market?

5.     Line 94-95: “The prices of crude glycerol continue to decrease because of the increasing biodiesel production and negatively impact the overall biodiesel production process”. How does the overall biodiesel production process is negatively impacted?

6.     Figure 1 provides the estimated production of crude glycerol in different countries up to the year 2020. What is the significance of this projection graph since the period 2000-2020 is already finished? The actual production quantity data may be available. The graph is not cited too. It seems the authors reused the graph from [23], which is published in 2012. The actual graph was taken from ‘Glycerine market analysis’ done in the year 2007. Reusing the projection graphs from the year 2007 does not make sense in this study.

7.     Line 115-116: “Hydrogen can also be used to produce energy production during its generation process itself after further development” – Consider rewriting the sentence. Check the usage of ‘produce energy production. What do you mean by ‘further development? Mention those processes, if any.

8.     Line 124-125: “Even though these processes are largely endothermic, they present high conversion and efficiency mates”: Seems the term ‘efficiency mates’ is a typographical error. Otherwise, please explain the term.

9.     Line 134-136: “They proposed the optimum temperature to be around 625oC and 1 bar, hydrogen production was maximized, whereas the side reactions were minimized.” Optimum pressure is included. So, revise the sentence.

10.  Line 137-139- “The POR method is governed by the glycerol oxidation reaction, where glycerol is reacted with oxygen at 1atm pressure to produce the same syngas as steam reforming. CO, CO2 and H2 [31]”. The second sentence is incomplete. English grammar usage and style should be improved in Section 3. Sentences should not make confusion to the readers.

11.  Section 4 lacks some introductory statements. Readers should understand which process the author is discussing. Also, the authors need to give an idea about the process integration and its necessity.

12.  Line 200-203: “The technique consists of a packed bed reactor and strippers which uses micro and mesoporous adsorbents to strip the gas, a simple version of this technique is simulated on Aspen Plus using the 'Sep' feature, a piece of equipment that separates an inlet stream based on the splits of each component required”. Split the sentence so that idea is conveyed easily.

13.  Line 205: “This equipment is of code B26 in the next image” – Instead of using ‘next image’, mention the figure title.

14.  Flow process diagram can be detailed more in the proper order of flow

15.  The title ‘Section 6’ seems misplaced. The results and discussion section is too small compared to Section 6. Section 6 can be a sub-section in the previous section. Assign proper subsections to Section 5 and make ‘Results and discussion’ section comprehensive.

16.  Line 242-243: “The analysis report showed that when varying reactor temperature with the hydrogen flow rate and mole fraction, that they are consistent and increase together” – Reconstruct the sentence.

17.  Figures 3,4,5 are named the same.  Specific titles can be given instead of ‘sensitivity analysis’.

18.  Line 269-271: “Compared with the results of recent literature it showed new and advanced technologies for biodiesel production from waste cooking oil,  this value is quite low [19]. - The cited reference paper is published in 2008. So, it cannot be regarded as recent literature. Moreover, the statement is not clear.

19.  Conclusion part is clear and good. Sections 5 and 6 need considerable improvement in writing style and content discussion.

Author Response

Dear Reviewers,

Thank you so much for taking time to review the paper again and providing your valuable feedback on the research paper titled “Technological Assessment on Steam Reforming Process of Crude Glycerol to Produce Hydrogen in an Integrated Waste Cooking Oil-based Biodiesel Production Scenario” for publication in your esteemed journal. 

Based on your comments and feedback received, we have made the requested changes in the manuscript highlighted in different colors.

 comments highlighted in yellow color 

Additionally, please see the below with authors comments for all the received feedback. 

We thank the reviewers for all their time and hope the changes made in the revised manuscript address all their concerns. 

Should you have any further questions or clarifications, we are happy to address and assist. 

Best Regards, 

Authors

Reviewer 2 Report

The manuscript entitled “Technological Assessment on Steam Reforming of Crude Glycerol to Produce Hydrogen in an Integrated Waste Cooking Oil-based Biodiesel Production Scenario” aims to present research based on creating an integrated process for biodiesel production and glycerol valorization and its technical analysis and feasibility for real-life scenarios. This topic is highly important, but the manuscript is not prepared well. First of all, it is very short and inconclusive. None of the sections is covered properly. There is no in-depth literature analysis, and there is no discussion at all. The manuscript should be expanded and improved in order to be published. The discussion needs to be scientific and critical. I have some comments with a major revision to strengthen the work may authors consider them to get the work better.

1. In the abstract not only one sentence is enough to describe literature and add references. Explain what was the output of that work. The introduction is too extensive. Authors should put more effort into the story of the introduction to narrow it down to the problem statement. For example, searching the literature, the following papers are suggested to be read and used: " Transesterification of waste cooking oil using Clay/CaO as a solid base catalyst "," Biodiesel production from waste cooking oil in a micro-sized reactor in the presence of cow bone-based KOH catalyst ", "Transesterification of waste cooking oil using clinoptilolite/industrial phosphoric waste as green and environmental catalysts "," Use of graphene-based materials as carriers of bioactive agents", and " Optimizing the Production of Biodiesel from Waste Cooking Oil Utilizing Industrial Waste‐ Derived MgO/CaO Catalysts".

2. Text has some grammatical typos.

3. References are suggested to come after the name of the author. If you want to put reference after the author keeps the style the same in the whole text.

4. The font and size of some parts of the text and formula are not unified and the same.

5. The quality of the figures and the tables are satisfactory and should be improved.

6. The numbering of paragraphs has some mistakes, check and fix them.

7. Line and paragraph spacing are not the same in the whole text. Why such careless mistakes?

Author Response

Dear Reviewers,

Thank you so much for taking time to review the paper again and providing your valuable feedback on the research paper titled “Technological Assessment on Steam Reforming Process of Crude Glycerol to Produce Hydrogen in an Integrated Waste Cooking Oil-based Biodiesel Production Scenario” for publication in your esteemed journal. 

Based on your comments and feedback received, we have made the requested changes in the manuscript highlighted in different colors.

Reviewer 2 comments highlighted in blue color

Additionally, please see the below with authors comments for all the received feedback. 

We thank the reviewers for all their time and hope the changes made in the revised manuscript address all their concerns. 

Should you have any further questions or clarifications, we are happy to address and assist. 

Best Regards, 

Authors

Reviewer 3 Report

Dear Respected Authors:

Thanks a lot for your efforts and good paper. My comments are as follow:

1- Try to add more details in the abstract regarding the problem, solution, techniques, and the results. 

2- The resolution of the figure must be improved

3- Table 2 is not clear, kindly improve it with one column for each stream.

4- Why you did not make mass and energy integration using Aspen plus using energy analyzer. It will enrich your paper.

6- Its required to make economic analysis to your process to make sure that this process is profitable or not. 

7- Kindly add more details and information in the results section.

8- Kindly add more recent References to your paper.

9- The conclusion part must be improved. 

Author Response

Dear Reviewers,

Thank you so much for taking time to review the paper again and providing your valuable feedback on the research paper titled “Technological Assessment on Steam Reforming Process of Crude Glycerol to Produce Hydrogen in an Integrated Waste Cooking Oil-based Biodiesel Production Scenario” for publication in your esteemed journal. 

Based on your comments and feedback received, we have made the requested changes in the manuscript highlighted in different colors.

 comments highlighted in green color

Additionally, please see the below with authors comments for all the received feedback. 

We thank the reviewers for all their time and hope the changes made in the revised manuscript address all their concerns. 

Should you have any further questions or clarifications, we are happy to address and assist. 

Best Regards, 

Authors

Round 2

Reviewer 2 Report

It can publish now.

Reviewer 3 Report

Dear Authors

Thanks for your reply. Good luck and wish you a soon publication.